# Transglutaminase 2 Regulates HSF1 Gene Expression in the Acute Phase of Fish Optic Nerve Regeneration

**DOI:** 10.3390/ijms25169078

**Published:** 2024-08-21

**Authors:** Kayo Sugitani, Takumi Mokuya, Yu Kanai, Yurina Takaya, Yuya Omori, Yoshiki Koriyama

**Affiliations:** 1Department of Clinical Laboratory Science, Graduate School of Medical Science, Kanazawa University, 5-11-80 Kodatsuno, Kanazawa 920-0942, Japan; 2Graduate School and Faculty of Pharmaceutical Sciences, Suzuka University of Medical Science, 3500-3 Minamitamagaki, Suzuka 513-8670, Japan; koriyama@suzuka-u.ac.jp

**Keywords:** TG2, HSF1, Yamanaka factors, Klf4, Oct4, Sox2, retina, optic nerve regeneration, zebrafish, cell survival

## Abstract

Fish retinal ganglion cells (RGCs) can regenerate after optic nerve lesions (ONLs). We previously reported that heat shock factor 1 (HSF1) and Yamanaka factors increased in the zebrafish retina 0.5–24 h after ONLs, and they led to cell survival and the transformation of neuro-stem cells. We also showed that retinoic acid (RA) signaling and transglutaminase 2 (TG2) were activated in the fish retina, performing neurite outgrowth 5–30 days after ONLs. In this study, we found that RA signaling and TG2 increased within 0.5 h in the zebrafish retina after ONLs. We examined their interaction with the TG2-specific morpholino and inhibitor due to the significantly close initiation time of TG2 and HSF1. The inhibition of TG2 led to the complete suppression of HSF1 expression. Furthermore, the results of a ChIP assay with an anti-TG2 antibody evidenced significant anti-TG2 immunoprecipitation of HSF1 genome DNA after ONLs. The inhibition of TG2 also suppressed Yamanaka factors’ gene expression. This rapid increase in TG2 expression occurred 30 min after the ONLs, and RA signaling occurred 15 min before this change. The present study demonstrates that TG2 regulates Yamanaka factors via HSF1 signals in the acute phase of fish optic nerve regeneration.

## 1. Introduction

In contrast to those in mammals, the neurons in the central nervous system (CNS) of adult fish can survive after nerve injury; additionally, their axons can regrow, and the CNS functions can fully recover [1,2,3,4,5]. The fish visual system has often been used as a model of CNS nerve regeneration [6,7,8,9,10]. It takes 4 months from optic nerve crush to fully recover visual function in fish [11,12]. As part of our study, we first subdivided this long regeneration process into three periods (early, middle, and late) after optic nerve lesions (ONLs) using modern neurobiological techniques [12]. The first stage involves the preparation for nerve regeneration at 0–5 days [13,14,15,16,17,18], while in the second stage, optic axon regrowth occurs at 5–30 days [19,20,21,22,23], and the third stage involves synaptic refinement in the brain at 1–4 months [12,24]. All three stages occur after ONLs. We researched regeneration-associated genes (RAGs) specifically activated in the early and middle stages of fish retina regeneration [12,13,14,15,16,17,18,19,20,24]. For the early stage, we were interested in the cell survival of the injured retinal ganglion cells (RGCs) and the transformation of RGCs with regeneration ability. We searched for RAGs as early as possible after ONLs for cell survival and found that the transcriptional factor heat shock factor 1 (HSF1) was expressed in the retina after ONLs [15,16,18]. HSF1 mRNA was expressed in all nuclear layers at 0.5 h and peaked at 6 h before declining at 72 h after ONLs [15,18]. As for the transformed neuro-stem cells from adult RGCs, we hypothesize that the Yamanaka factors well known for their role as iPS cell producers, namely Oct4, Sox2, and Klf4 (OSK) [25,26,27,28,29], are specifically activated in injured RGCs shortly after ONLs. We showed that OSK mRNAs started to increase at 1 h and peaked in all nuclear layers of the zebrafish retina at 1–6 h after ONLs [18], and they were regulated by the HSF1 gene via the pretreatment of HSF1 morpholino and ChIP assay with anti-HSF1 antibodies [18]. We concluded that the RGCs of injured fish can survive and become immature neural cells, such as neuro-stem cells, through the activation of Yamanaka factors via HSF1 signaling shortly after ONLs within 1–2 h [18].

In the middle neurite outgrowth stage, transglutaminase 2 (TG2) mRNA increased in the RGCs at 5 days, peaked at 20 days, and declined at 40 days after ONLs [12,20]. A previous study revealed that the recombinant TG2 protein induced outgrowth in the explant culture of goldfish primed retina, leading to very long and thick neurites, and reported a significant neurite outgrowth in the explant culture of adult rat retina [20]. From these results, we conclude that TG2 is a strong potential contributor to neurite outgrowth and elongation toward the target tectum. In addition, retinoid-metabolizing enzymes and binding proteins are activated as retinoic acid (RA) signaling molecules for fish optic regeneration in this neurite outgrowth period [30]. For example, retinaldehyde dehydrogenase 2 (RALDH2) (a biosynthetic enzyme of RA), purpurin (a retinol-binding protein), and retinoic acid receptor αa (RARαa) are mRNAs that increased in RGCs at 5–10 days and declined at 40 days after ONLs. However, the mRNA of cytochrome P450 family 26 subfamily a polypeptide 1 (CYP26a1) is only reduced at 3–5 days in RGCs, reaching the maximum reduction at 5–7 days and returning to the control by 40 days [30]. RA is a major transcription factor of TG2 [31,32,33,34,35], suggesting that TG2 is regulated via RA signaling during the middle stage of regeneration after ONLs.

Previous studies have shown that Yamanaka factors are expressed spontaneously within 1 h after ONLs in the retina of zebrafish [18]. Here, we demonstrate for the first time that, through RA signaling, TG2 is the most rapidly activated gene in the fish retina after ONLs, with its initiation occurring within less than 15 min, and it directly regulates Yamanaka factors via HSF1 signaling.

## 2. Results

### 2.1. A Rapid Increase in Transglutaminase 2 (TG2) Immediately after ONLs

We performed real-time PCR using gene-specific primers to examine how TG2 gene expression changes in the retina after optic nerve crush (see Table 1). The upregulation of TG2 mRNA peaked at 0.5 h and then decreased to control levels 6–24 h after ONLs (Figure 1a). The expression levels of these TG2 mRNAs were found in all the nuclear layers of the retina via in situ hybridization (Figure 1b). However, no positive signals were observed with the sense probe (Sense). Through immunohistochemical staining analysis, we also confirmed that the time course of TG2 protein expression was the same as that of other factors after ONLs (Figure 1c).

### 2.2. A Rapid Increase in Retinoic Acid (RA) Signaling Immediately after ONLs

TG2 expression is regulated through various biological events, and retinoic acid (RA) is one of the well-known inducers. We investigated the potential role of RA as a rapid increase in TG2 expression was observed in the retina after ONLs. We analyzed the changes in the following four factors: (1) the retinoic acid receptor alpha a (RARαa); (2) retinaldehyde dehydrogenase 2 (RALDH2), which is involved in the production of retinoic acid from retinal; (3) purpurin, a retinol-binding protein; and (4) cytochrome P450 26a1 (CYP26a1), which contributes to the metabolization of RA to its inactive form. The results showed that the expression of RARα, RALDH2, and purpurin mRNAs increased as rapidly as TG2 within 0.25–0.5 h after the ONLs (Figure 2a–c). On the other hand, CYP26al expression was rapidly downregulated 3 h after the ONLs (Figure 2d). These results suggest that the injured retina promotes a response to generate RA, and the increase in RA is strongly associated with the upregulation of TG2 expression.

### 2.3. TG2 Regulates the Expression of HSF1 after ONLs

Next, we investigated the relationship between TG2 and HSF1, as HSF1 has been known as the most rapidly changing molecule after ONLs to date. We used each specific morpholino (MO) for suppression and examined the changes in TG2 or HSF1 expression profiles. Our results showed that TG2 mRNA expression was markedly suppressed after TG2-specific MO injection into the eye (Figure 3a). The measurement of HSF1 mRNA expression in these samples showed similarly marked decreases (TG2 MO, Figure 3b). In contrast, the standard MO injection samples showed a marked increase in HSF1 mRNA expression 1 h after the ONLs (Std MO, Figure 3b). HSF1 mRNA expression was suppressed after HSF1-specific MO injection into the eye (Figure 3c). However, the measurement of TG2 mRNA expression in these samples increased similarly to that in the standard MO injection samples (Figure 3d). These results indicate that HSF1 mRNA expression does not affect TG2 mRNA expression.

### 2.4. TG2-Specific Inhibitor Suppressed HSF1 Expression in the Retina after ONLs

We performed an ocular injection of a TG2-specific inhibitor Z-DON (1 mM) 1 h before the ONLs to confirm the regulation of HSF1 expression via TG2. Under these conditions, the retina was removed 0.5 h after the ONLs, and retinal flat mounts were prepared for reaction with the FITC-labeled TG2-specific substrate. TG2 expression after the ONLs was markedly suppressed at the enzymatic protein level (Figure 4a, TG2 inhibitor, Figure 4b). Compared with the control DMSO dilution injection samples (Figure 4a, DMSO 0.5 h), the TG2 inhibitor-treated samples had almost no enzyme activity; thus, the response of the FITC-labeled TG2-specific substrate could hardly be observed in the flat-mount retina (Figure 4a,b). Under these conditions, HSF1 mRNA expression was also significantly suppressed (Figure 4c).

### 2.5. ChIP Assay for HSF1 in Response to TG2

To confirm the association between the induction of TG2 expression after ONLs and the subsequent increase in HSF1, we carried out a ChIP assay, an experimental method for detecting protein–DNA interactions. Thus, after the cross-linking reaction of TG2 protein and HSF1 DNA bonding with formaldehyde in the retinal samples, immunoprecipitation was performed using an anti-TG2 antibody, and the coprecipitated HSF1 DNA was subjected to PCR. DNA samples were collected from the intact retina (0 h) or injured retina (0.5 h) after ONLs. These DNA samples were immunoprecipitated with the anti-TG2 antibody. After DNA purification, the ChIP-enriched samples were amplified with several primer sets encoding HSF1. As a result, the anti-TG2 antibody specifically precipitated HSF1 genomic DNA approximately 40–50 times more than the IgG control (Figure 5; 0.5 h). Almost no amplified products were detected in the IgG control-treated samples (Figure 5, IgG) or the intact control samples (Figure 5, 0 h).

### 2.6. Effect of TG2 Morpholino (MO) on Yamanaka Factor Expression after ONLs

Previous studies have demonstrated that HSF1 is involved in the induction of Yamanaka factor expression in the retina after ONLs [18]. The intraocular injection of HSF1-specific MO before the ONLs led to the complete suppression of not only the increase in HSF1 mRNA expression in the retina 6 h after the ONLs but also the upregulation of Yamanaka factors. Therefore, the inhibition of TG2 expression leads to the suppression of Yamanaka factors’ expression via the downregulation of HSF1. Consequently, we investigated the expression profiles of three Yamanaka factors (klf4, oct4, and sox2) with the suppression of TG2 expression through MO treatment. The results revealed that, through treatment with TG2 MO, the expression of these genes was also markedly suppressed 1 h after the ONLs compared with the control, standard MO injection group (Figure 6a–c).

These results strongly suggest that TG2 has an HSF1-mediated regulatory mechanism for expressing the Yamanaka factors after ONLs. Further study is needed to investigate whether TG2 is directly involved in the expression of Yamanaka factors.

### 2.7. Effect of TG2 Morpholino (MO) on Apoptosis in the Retina after ONLs

In a previous study, the inhibition of HSF1 expression after ONLs using HSF-1-specific MO induced apoptosis in the retina within 24 h [18]. In this study, TG2 expression increased faster than HSF1 after the ONLs, suggesting that TG2 may regulate HSF1 during apoptosis in the retina. To investigate the potential role of TG2 in cell survival after ONLs, TG2-specific MO was injected intraocularly before optic nerve injury to examine its effects. The injection of TG2-MO alone did not increase apoptosis-positive cells (the intact optic nerve in Figure 7). Few apoptosis-positive cells were observed in optic nerve crush and Std MO groups. Only in the experimental group subjected to TG2-MO injection and optic nerve crush was the structure of the retinal layer almost fully destroyed, and numerous apoptotic cells were observed within 24 h.

## 3. Discussion

### 3.1. Rapid Activation of RA Signaling and TG2 Gene Expression in the Zebrafish Retina after ONLs

In light of our previous results indicating that TG2 plays a pivotal role in optic axon regrowth through RA signaling during the middle stage of fish optic nerve regeneration [30], we scrutinized for genes activated in the zebrafish retina in the acute phase immediately after ONLs. Increases in RA signaling at 15 min and TG2 at 30 min in the retina were also identified after the ONLs. Our results prove that RA signaling, the upregulation of RALDH and purpurin, and the downregulation of CYP26a1 all lead to an increase in RA biosynthesis. The upregulation of RARα leads to the acceleration of intranuclear transcription [32,33,36,37]. Purpurin increased in the photoreceptor 3–5 days after the ONLs and, together with retinol, facilitated neurite outgrowth [13,17]. In our research, we considered purpurin to be an early gene activated 2–3 days after the ONLs [13,17]. It is a secretory protein and acts in RA biosynthesis through retinol–retinal conversion from the photoreceptor to all other retinal cells [13,17,30,38]. This increased expression of RA signaling may also have a significant effect on various cell survival mechanisms, as all cell types express nuclear RA receptors (RA nuclear receptors and retinoid X receptors), which may positively or negatively regulate the expression of RA target genes in response to RA [12,33,34]. The diverse factors involved in the regulation of target gene expression are believed to be the result of the complexity of signaling pathways at various levels, for example, due to the existence of two families of RA receptors, three RAR isotypes (α, β, and γ) and three RXR isotypes (α, β, and γ) and their combination as heterodimers [39,40]. The 5′ upstream promoter region of the TG2 gene contains a retinoic acid-binding region with RAR and RXR heterodimers [41], which are likely involved in the increased expression of the TG2 gene after optic nerve lesions in zebrafish. During the optic nerve regeneration of zebrafish, a bimodal upregulation of TG2 expression occurs in both the acute and middle phases (optic axon regrowth) after ONLs. We found that the upregulation of retinal transglutaminase (TGR), which has a high homology to zebrafish TG2b, was a strong promoter of neurite outgrowth [20].

In addition to retinoic acid signaling, TG2 expression is regulated by various molecules, such as transforming growth factor (TGF)-β [42], interleukin (IL)-1 [43], IL-6 [44], tumor necrosis factor (TNF)-α [45], hypoxia-inducible factor-1 [46], epidermal growth factor (EGF) [47], etc. Other factors may also be involved in the zebrafish optic nerve injury model. Thus, TG2 has many physiological functions [32,33,48,49,50,51] that contribute to maintaining homeostasis in various biological events.

### 3.2. TG2-Activated HSF1 in the Zebrafish Retina in the Acute Phase after ONLs

We have thus far established that HSF1 is the earliest gene activated in the fish retina after ONLs [15,16,18]. However, the present study clearly shows that RA signaling and TG2 increased in the retina more rapidly after the ONLs. We also found that TG2 regulated HSF1 gene expression using TG2-specific MO and ChIP assay with an anti-TG2 antibody. We previously described that HSF1 is one of the most powerful candidate genes for cell survival in injured fish retina after ONLs [18]. However, in this study, TG2 was found to respond faster than HSF1 and contribute to cell survival after the ONLs (Figure 7). Piacentini et al. reported that TG2 regulates HSF1 through post-translational modification resulting from the protein disulfide isomerase activity [52]. Furthermore, TG2-specific MO led to the complete suppression of the gene expression of Yamanaka factors, which are the key molecules involved in the survival of injured RGCs and the transformation of neuro-stem cells in the early stage of optic nerve regeneration in fish through their activation via HSF1. It was reported that when the three Yamanaka factors Klf4, Oct4, and Sox2 were overexpressed by a viral vector, part of the optic nerve was regenerated, and visual function was partially restored [53]. Our mouse optic nerve crush experiments revealed no increase in HSF1 expression in the retina after the ONLs (see Appendix A). Furthermore, TG2 expression completely disappeared in the retina of rats 7 days after ONLs [20]. At present, we do not know the exact molecular mechanism underlying this occurrence, so the TG2 substrate should be investigated as a protein cross-linking enzyme. This study presents the experimental results of TG2 downregulation using a single MO and a TG2-specific inhibitor. Additional studies are needed to confirm the effects of other MO treatments [54,55] and gene modification using CRISPR-based gene editing [56] on the association between TG2 expression in the retina and the cascade response to optic nerve regeneration. Nevertheless, TG2 regulates HSF1 and the gene expression of Yamanaka factors, thus contributing to maintaining cell survival and improving the regenerative ability of the injured RGCs in the fish retina immediately after ONLs.

## 4. Materials and Methods

### 4.1. Ethics Statement

All the experimental procedures were approved by the Committee on Animal Experimentation of Kanazawa University, approval number # AP24-006, on 4 April 2024. All animal care was performed in accordance with the guidelines for animal experiments of Kanazawa University, and special care was taken to minimize the suffering of the fish.

### 4.2. Animals

Adult zebrafish (Danio rerio; 3–4 cm in length) were used throughout this study. The zebrafish were anesthetized with 0.02% MS222 (Sigma-Aldrich, St. Louis, MO, USA) in 10 mM phosphate-buffered saline (PBS; pH 7.4). Under anesthesia, the optic nerve on both sides was carefully crushed with forceps 1 mm posterior to the eyeball. Following surgery, the fish were maintained in water tanks at 28 °C until the appropriate time points.

### 4.3. Tissue Preparation

Retinal tissues were collected for histological analysis at specific time points following the ONLs. After the eye had been enucleated, the lens was removed and fixed in 4% paraformaldehyde solution containing 0.1 M phosphate buffer (pH 7.4) and 5% sucrose for 2 h at 4 °C. The eyes were infiltrated with increasing concentrations of sucrose from 5% to 20%, followed by overnight incubation in 20% sucrose at 4 °C. The tissues were embedded in an optimal cutting temperature (OCT) compound (Sakura Fine Technical, Tokyo, Japan), and thin sections with 12–14 μm thickness were prepared.

### 4.4. Total RNA Extraction for cDNA Synthesis

Zebrafish were euthanized with an overdose (0.1%) of MS222 in PBS, followed by exposure to ice-cold water at appropriate time points after the ONLs. Total RNA was prepared using Isogen (Nippon Gene, Tokyo, Japan) according to the manufacturer’s instructions, and the samples were subjected to first-strand cDNA synthesis using a Transcriptor High-Fidelity cDNA Synthesis Kit (Roche, Mannheim, Germany).

### 4.5. Quantitative Real-Time PCR

We performed quantitative real-time PCR with Power SYBR Green PCR Master Mix (Thermo Fisher Scientific, Waltham, MA, USA) using a QuantStudio 3 real-time system (Thermo Fisher Scientific). Based on the zebrafish cDNA sequences, we created gene-specific primers using Primer3 (version 0.4.0) and BLAST (Table 1). We analyzed the expression levels via the ΔΔCt method. Glyceraldehyde 3-phosphate dehydrogenase (GAPDH) was used as a reference gene. The accession numbers for the genes and DNA sequences of the primer pairs used in each experiment are shown in Table 1.

### 4.6. Immunohistochemistry

Retinal sections were incubated at 121 °C for 10 min in 10 mM citrate buffer. Following washing and blocking, sections were incubated with primary antibodies overnight at 4 °C (anti-TG2, 1:500, CUB7402 Neomarkers, Fremont, CA, USA). Following incubation with a biotinylated secondary antibody (Vector Laboratories, Newark, CA, USA) for 2 h at room temperature, the bound antibodies were detected using horseradish peroxidase (HRP)-conjugated streptavidin and 3-amino-9-ethyl carbazole (AEC; Nichirei Biosciences Inc., Tokyo, Japan).

### 4.7. In Situ Hybridization

We carried out in situ hybridization as previously described [20]. Briefly, retinal sections were rehydrated and treated with 5 mg/mL proteinase K (Invitrogen, Carlsbad, CA, USA) at 22 °C for 5 min. Following acetylation and prehybridization, hybridization was performed overnight at 42 °C using digoxigenin-labeled cRNA probes. The next day, the sections were washed and treated with RNase A at 37 °C for 30 min to completely remove the free cRNA probes. The sections were incubated with an alkaline phosphatase-conjugated anti-digoxigenin antibody (Roche, Rotkreuz, Switzerland) overnight at 4 °C to detect the signals. Tetrazolium-bromo-4-chloro-3-indolylphosphate (Roche) was used as the substrate to visualize positive signals.

### 4.8. Chromatin Immunoprecipitation

We performed chromatin immunoprecipitation (ChIP) using the MAGnifity Chromatin Immunoprecipitation System (Thermo Fisher Scientific, Waltham, MA, USA) according to the manufacturer’s instructions. Briefly, we homogenized the retinal samples and linked them in 1% formaldehyde for 10 min at room temperature, following the addition of 100 mM glycine to stop the reaction and washing with cold PBS three times. After centrifugation and ultrasonication using a Bioruptor ultrasonic homogenizer (BM Equipment Co., Ltd., Tokyo, Japan), we incubated the samples with magnetic protein A/G beads conjugated with anti-TG2 [CUB 7402] (Abcam, Waltham, MA, USA) or normal IgG and left them overnight at 4 °C. Following immunoprecipitation and washing, we purified the genomic DNA associated with TG2 and analyzed it using SYBR Green-based quantitative real-time PCR with Power SYBR Green PCR Master Mix (Thermo Fisher Scientific, Waltham, MA, USA). A ChIP dilution buffer was used as a negative control, and DNA from the total input was used as an internal positive control.

### 4.9. Intraocular Injection of TG2 or HSF1 Morpholino into Zebrafish Eye

Vivo-morpholino (MO) for transglutaminase 2 (TG2) was prepared to inhibit zebrafish *TGM2* expression using the following sequence: 5′-CATGAGCCGATGTCCAGAGCCAT-3′. This sequence is complementary to exons 1-exon 2 of *Danio rerio* TG2b mRNA. TG2 MO treatment significantly suppresses TG2b expression compared with standard (Std) MO-treated samples (see Appendix A). In addition, HSF1-MO was used to suppress heat shock factor 1 (HSF1) expression with the following sequence: 5′-AGTTTAGTGATGATTTCTGACGGTA-3′. This HSF1-MO has complementary sequences to the HSF1 mRNA 5′-UTR in exon 1. We used a standard vivo-MO (5′-CCTCTTACCTCAGTTACAATTTATA-3′) as a control. All the MOs were purchased from GeneTools (OR, USA). A 0.5 µL volume of MO solution (0.5 mM) was injected into the eye with a Hamilton 33G neuron syringe. The optic nerve was then crushed twenty hours after the MO treatment. One hour after the ONLs, retinal samples were removed, and total RNA was extracted.

### 4.10. Intraocular Injection of TG2-Specific Inhibitor into Zebrafish Eye

Z-DON (ZEDIRA GmbH, Darmstadt, Germany, gift from Prof. Hitomi, Nagoya univ.), a TG2-specific membrane-permeable blocker, was used for the TG2 inhibition test. A total of 100 mM Z-DON in DMSO (dimethyl sulfoxide) was diluted to 1 mM in PBS, and 0.5 µL was injected into both eyes with a Hamilton 33G neuron syringe 1 h before the ONLs. Immediately after the ONLs, the optic nerve crush site was treated with an additional 1 µL of 1 mM Z-DON in PBS. The diluted DMSO solution with PBS was used as the negative control. Retinal samples were removed 0.5 h after the ONLs, and TG enzyme activity assay (next section) and real-time PCR were performed.

### 4.11. In Situ TG Activity Assay with Flat-Mount Retina after Treatment of TG Inhibitor

For the TG enzyme assay, retinal tissue was prepared without fixation to the flat-mount assay using FITC-labeled TG2-specific substrates (gift from Prof. Hitomi, Nagoya univ.). Briefly, retinal samples were removed and washed three times with PBS in a 96-well microplate (stem, Tokyo, Japan). After Blocking (1%BSA in PBS) and washing, flat-mount retinas were incubated with 1 µM FITC-conjugated peptide pepT2 (HQSYVDPWMLDH), which has a specific affinity for TG2 [57] in 200 µL of a substrate reaction solution (100 mM Tris-HCl pH 8.0, 1 mM CaCl_2_, and 1 mM DTT) for 2 h at 28 °C. To avoid non-specific reactions, the mutant peptide pepT26QN, in which the reactive glutamine residue was replaced with asparagine, was used as a negative control. Following the stop reaction with 25 mM EDTA in PBS and washing, the bound FITC signals were detected using a fluorescence microscope (E600; Nikon, Tokyo, Japan). TG enzymatic activity was quantified via the analysis of fluorescence intensity versus control (0 h). Data are expressed as the mean ± SEM of five independent experiments and analyzed by one-way ANOVA, followed by Scheffe’s multiple-comparison test.

### 4.12. Terminal Transferase-Mediated dUTP Nick-End Labeling (TUNEL) Staining

For the detection of apoptotic cells, we performed TUNEL staining using an In situ Apoptosis Detection Kit (TaKaRa, Shiga, Japan) according to the manufacturer’s instructions. Briefly, retinal sections were treated with 0.3% H_2_O_2_ for 20 min after PBS washing. After treatment with a permeabilization buffer, the sections were reacted with a labeling mix solution containing terminal transferase and fluorescence dUTP for 70 min at 37 °C. After this step, samples were observed using a fluorescence microscope. For observation under visible light, the sections were reacted with biotinylated anti-FITC (1:100, SouthernBiotech, Birmingham, AL, USA), HRP–streptavidin (Nichirei Bioscience, Tokyo, Japan), andAEC substrate solution (Nichirei Bioscience, Tokyo, Japan).

### 4.13. Statistical Analysis

The expression levels of TG2, RALDH2, RARαa, CYP26a1, and HSF1 are expressed as mean ± SEM. Differences in mRNA expression were evaluated via one-way ANOVA, with *p* < 0.05 indicating a significant difference using IBM SPSS Statistic software (29.0.0).

## Figures and Tables

**Figure 1 ijms-25-09078-f001:**
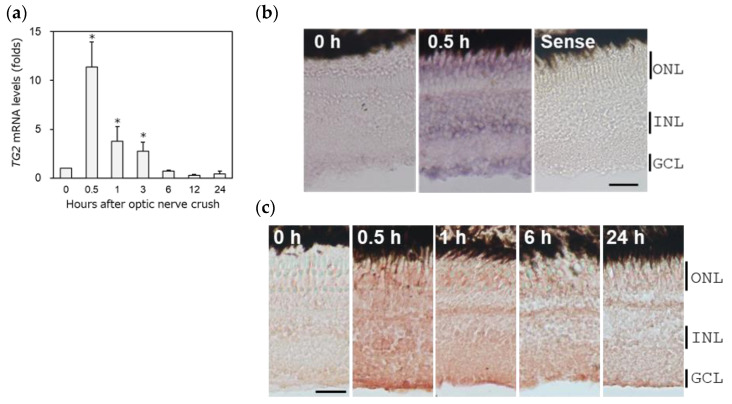
The upregulation of TG2 (transglutaminase 2) in the zebrafish retina after ONLs (optic nerve lesions): (**a**) TG2 mRNA expression levels were determined using quantitative real-time PCR. Data are expressed as the mean ± SEM, with statistical significance set at * *p* < 0.05. (**b**) The in situ hybridization of TG2 in zebrafish retina. TG2 mRNA peaked in the retina 0.5 h after the ONLs; its localization was observed in all nuclear layers in the retina, i.e., the ONLs (outer nuclear layers), the INLs (inner nuclear layers), and the GCLs (ganglion cell layers). (**c**) The immunohistochemical staining of TG2 in the zebrafish retina. Significant immunostaining was observed at 0.5 h after the ONLs in all nuclear layers, particularly in the GCLs. Scale bar = 20 μm.

**Figure 2 ijms-25-09078-f002:**
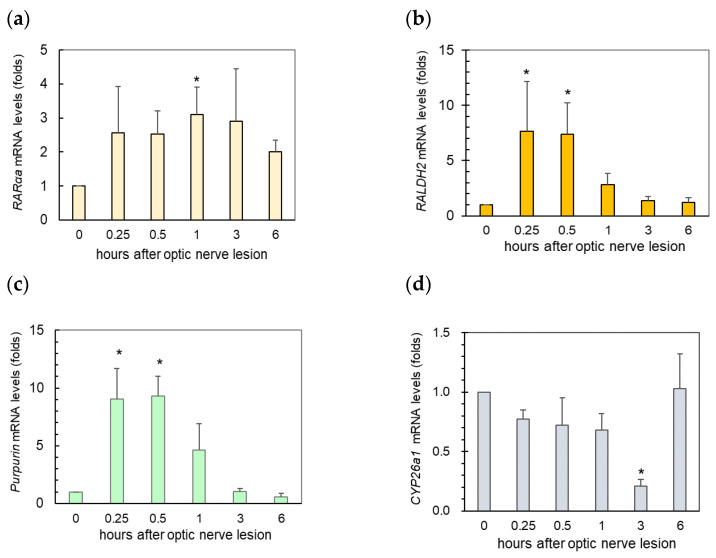
The upregulation of the RA signaling pathway in the zebrafish retina immediately after optic nerve lesions (ONLs): (**a**) RARαa, (**b**) RALDH2, and (**c**) purpurin were upregulated in the retina within 0.25–1 h, while (**d**) CYP26a1 was downregulated 3 h after the ONLs. The experiments were repeated 6–7 times. Statistical analysis was performed using one-way ANOVA, followed by Scheffe’s multiple-comparison test. Data are expressed as the mean ± SEM, with statistical significance set at * *p* < 0.05.

**Figure 3 ijms-25-09078-f003:**
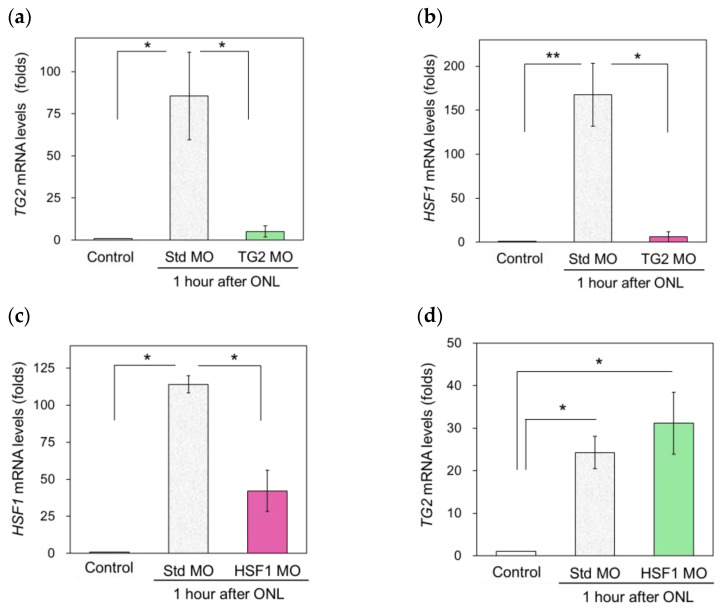
Injection of TG2 MO (morpholino) significantly reduced HSF1 mRNA expression 1 h after ONLs: (**a**) The TG2 MO-treated samples exhibited significant suppression of TG2 mRNA expression compared with the Std. MO-treated samples. (**b**) Under these TG2 inhibitory conditions, HSF1 mRNA expression was also inhibited compared with that of the control (Std. MO) samples. (**c**) The HSF1 MO-treated group exhibited suppression of HSF1 mRNA expression 1 h after the ONLs compared with the Std. MO-treated samples. (**d**) Even under these HSF1 inhibitory conditions, TG2 mRNA expression was unaffected and increased compared with the control samples rather than tending to be higher than in the Std. MO-treated samples. The experiments were repeated five to six times. Statistical analysis was performed using one-way ANOVA, followed by Scheffe’s multiple-comparison test. Data are expressed as the mean ± SEM, with statistical significance set at * *p* < 0.05, ** *p* < 0.01.

**Figure 4 ijms-25-09078-f004:**
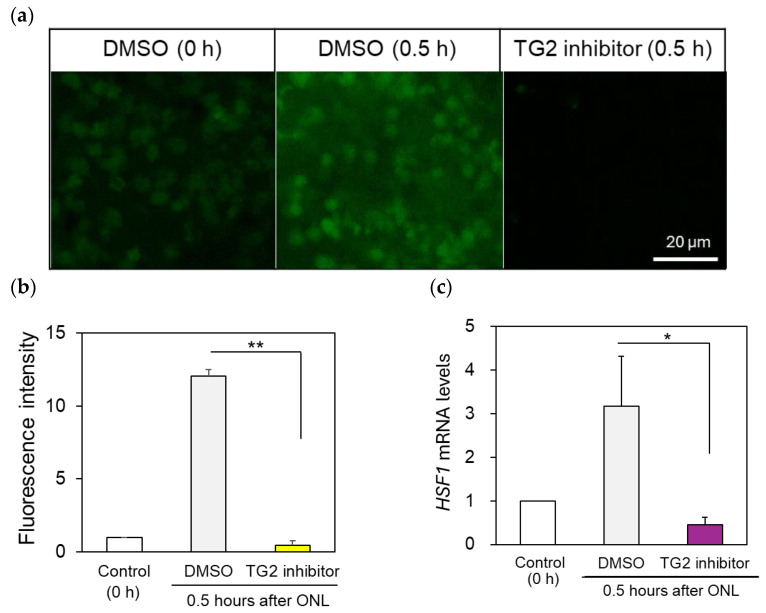
Treatment with a TG2-specific inhibitor, Z-DON (1 mM), reduced TG enzyme activity 0.5 h after the ONLs and significantly decreased HSF1 mRNA expression: (**a**) TG2 enzymatic activity was clearly suppressed in the flat-mount retina with Z-DON-injected samples compared with the DMSO injected samples (DMSO 0.5 h). (**b**) FITC-labeled TG2-specific substrates exhibited almost no response in the TG2 inhibitor-treated retina. (**c**) TG2-specific inhibitor-treated samples exhibited suppression of HSF1 mRNA expression compared with the DMSO-treated samples. The experiments were repeated five to six times. Statistical analysis was performed using one-way ANOVA, followed by Scheffe’s multiple-comparison test. Data are expressed as the mean ± SEM, with significance set at * *p* < 0.05 and ** *p* < 0.01. Bar, 20 µm.

**Figure 5 ijms-25-09078-f005:**
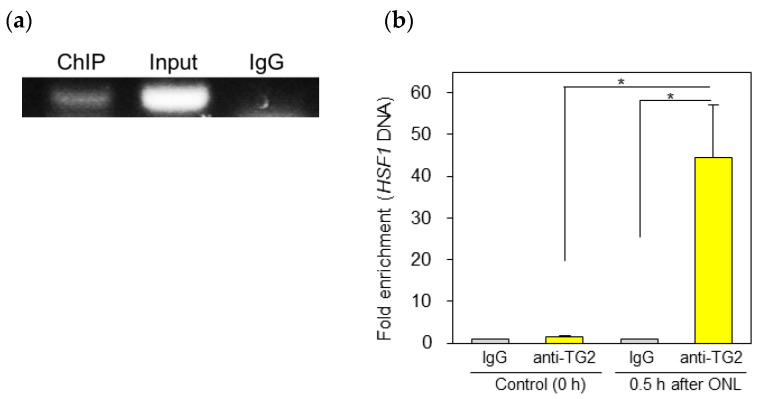
The results of the ChIP assay revealed the binding of anti-TG2 antibodies to the HSF1 genomic DNA. ChIP-enriched DNA was immunoprecipitated using IgG or anti-TG2 antibody from the control (0 h) or damaged zebrafish retina 0.5 h after ONLs: (**a**) Gel electrophoresis image for the ChIP samples. The input was an internal positive control for the ChIP assay. (**b**) The immunoprecipitated HSF1 DNA was analyzed using real-time PCR. Each ChIP signal was divided by the control IgG signals, and the results are presented as a fold increase in the signal relative to the background signal. Statistical analysis was performed via one-way ANOVA, followed by Scheffe’s multiple-comparison test. Data are expressed as the mean ± SEM of 5–6 independent experiments, with statistical significance set at * *p* < 0.05.

**Figure 6 ijms-25-09078-f006:**
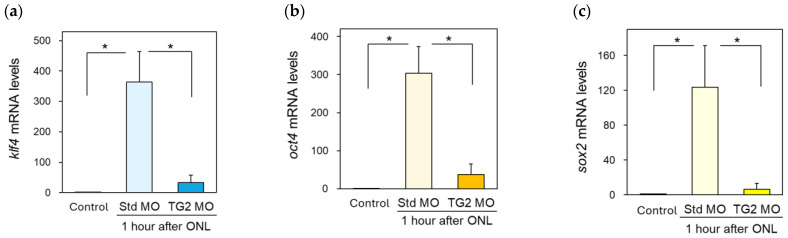
Treatment with TG2 MO (morpholino) significantly downregulated the mRNA expression of Klf4, Oct4, and Sox2 1 h after ONLs. TG2 MO or standard MO (Std. MO) was injected intraocularly 20 h before the ONLs. The TG2 MO-treated samples exhibited suppression of the mRNA expression of klf4 (**a**), oct4 (**b**), and sox2 (**c**) compared with that in the control (Std. MO) samples. Statistical analysis was performed using one-way ANOVA, followed by Scheffe’s multiple-comparison test. Data are expressed as the mean ± SEM of five to six independent experiments, with statistical significance set at * *p* < 0.05.

**Figure 7 ijms-25-09078-f007:**
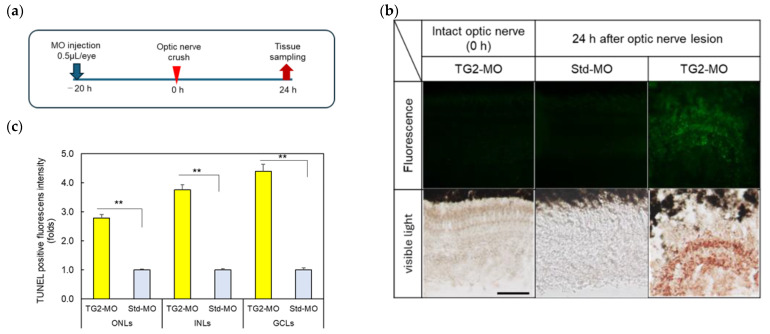
The inhibition of TG2 expression induced neuronal apoptosis in the retina after optic nerve lesions: (**a**) Timeframe for the administration of TG2-MO (morpholino) or standard MO (Std-MO), the occurrence of optic nerve lesion, and sample collection. (**b**) The detection of apoptotic cells in the retina 24 h after ONLs through MO administration. Optic nerve crush after the intraocular injection of TG2-specific MO resulted in a marked disruption of the structure of the retinal layer and a significant increase in apoptosis TUNEL-positive cells (FITC fluorescence-positive cells in the upper panel and the red-colored cells in the lower panel). Scale bar = 50 μm. (**c**) Apoptotic cells were quantified by analyzing fluorescence intensity versus Std-MO control. Data are expressed as the mean ± SEM of five independent experiments and analyzed using one-way ANOVA, followed by Scheffe’s multiple-comparison test. Statistical significance was set at *** p* < 0.01. ONLs, outer nuclear layers; INLs, inner nuclear layers; GCLs, ganglion cell layers.

**Table 1 ijms-25-09078-t001:** Sequences of the PCR primers used in this study.

Gene	Accession No.	5′ Primer	3′ Primer	Purpose
*HSF1*	NM_001313736	GATCTGCTGGAGCCCAAA	TCGGCAGAACTTCTTTGGAA	real-time PCR
		GGAGCTCCAGGATGACTCGT	GAACAGGCTGGAAGTTGAGC	ChIP assay
*TG2 (TG2b)*	NM_212656	TCCAGTCCACCAGGAGAATC	TGGATCCGCTCGTCTAGAGT	real-time PCR
		GCCCGGAACAGCAGATCA	AACATACTCCGCCAGCTCT	in situ hybridization
*Oct4*	NM_131112	CAACTCCCTCCGCTTCATC	GCTTCCGAACCCATTTCC	real-time PCR
*Sox2*	NM_213118	GACCATTCATCGACGAAGCC	CCTCCGGGGTCTGTATTTGT	real-time PCR
*Klf4*	NM_001113483	ACCGATGTGAAGCACAAGG	GCAGGTCGCACCTGTAGAC	real-time PCR
*RALDH2*	NM_131850.1	ACAGTGCTTACCTTGCTACCC	CTTATCTGCCCATCCAGCGT	real-time PCR
*CYP26a1*	NM_131146.2	TCAGGGTGATGGGAGCTGAT	GTCAGAGCCCAGGATGGTTC	real-time PCR
*RARαa*	NM_131406.2	TGATTAAACCCGCGTCTGTG	AGCTCCGGTTATTTAGTCTCGT	real-time PCR
*GAPDH*	NM_001115114.1	TCAGTCCACTCACACCAAGTG	CGACCGAATCCGTTAATACC	real-time PCR

## Data Availability

The data presented in this study are openly available at https://kanazawa-u.repo.nii.ac.jp/?page=1&size=20&sort=custom_sort&search_type=0&q=0.

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
