# Peer review of "Transglutaminase 2 Regulates HSF1 Gene Expression in the Acute Phase of Fish Optic Nerve Regeneration"

_ijms, 2024, doi:10.3390/ijms25169078_

Round 1

Reviewer 1 Report

Comments and Suggestions for Authors

The manuscript by Sugitani et al., presents an interesting story that demonstrates that TG2 directly regulates HSF1 gene expression that is required for optic nerve regeneration after a crushing injury. This builds on their previous reports that show that HSF1 regulates OCT4/SOX2/Klf4 expression that is required for RGC repopulation/outgrowth after injury. There are several major issues with the manuscript however that need to be addressed before publication. 

1)        Morpholino controls- While the authors use a standard negative control  morpholino, and demonstrate reduced tg2/hsf1expression in their morphants, critical information is lacking. It is stated that the morpholinos used are designed to reduce expression of tg2/hsf1. Can the authors elaborate? I assume these must be spice blocking morpholinos, and a translation blocking morpholino wouldn’t affect expression levels. Presumably, the inclusion of intron sequence, or the exclusion of exon sequence if a cryptic splice site is activated could lead to nonsense mediated decay and reduced expression. The authors at the very least should perform a PCR/gel based assay to demonstrate the incorrect splicing of the transcript after morpholino injection. Bioinformatic tools could also be used to predict the consequences of the morpholino on splicing. Repeating the results with a non-overlapping second morpholino, or in an F0 Crispant would be much more convincing.

2)        Retinoic acid signalling- while the authors show changes in expression of RA related genes after optic nerve injury, any role in regulating RGC survival or optic nerve regeneration in merely speculative. No manipulations of RA signalling were conducted, so it is difficult to state that thes changes in expression contribute to  downstream events such as the regulation of TG2 or HSF1. Thus the role of RA in this process is overstated throughout the manuscript. The use of DEAB, an inhibitor of raldh2 (aldh1a2), or RA itself, could be used with subsequent testing of tg2/hsp1 transcript or protein levels. This would demonstrate the role of RA in response to injury. 

3)        Tg2 enzymatic assay (Figure 4)- No description of the assay is given. In the TG2 inhibitor panel of figure 4A, cells are not visible. This would be more convincing with a nuclear stain included, so we can see the cells in the inhibitor panel. Also, the picture is very zoomed in. It is difficult to tell what part of the eye is photographed. The RGC layer in adults is very thin (usually just a couple of cells layers thick) so this picture (control panel) is likely not of the ganglion cell layer. A picture with a sider filed of view, with a counter nuclear stain should be added to the paper, along with a description of the enzymatic assay. 

4)        No negative control (no primary antibody control) was included for the immunohistochemistry. This is particularly important in the supplemental Figure, where the staining for all panels is weak. Some panels look a little darker than others, but it is difficult to interpret without the negative control. 

5)        The discussion is very brief, and just restates the results. Many papers from this field are not referenced, nor is there any description of how this work affects the field overall. Given the authors state that there is no increased HSF1 in the mouse retina after injury (supplemental Figure), but there is in the fish, differences with other animal models should be discussed. 

Comments on the Quality of English Language

Minor editing required.

Author Response

Dear Reviewer 1,

Thank you very much for your editorial work on our manuscript entitled “Transglutaminase 2 Regulates HSF1 Gene Expression in the Acute Phase of Fish Optic Nerve Regeneration” (Manuscript ID: ijms-3105568). 

We have revised it according to your comments and suggestions. The followings are the specific suggestions and our replies.

1-a) Morpholino controls- While the authors use a standard negative control morpholino, and demonstrate reduced tg2/hsf1expression in their morphants, critical information is lacking.

-> The standard morpholino, used as control MO, consists of 25 nucleotides, there is no complementation in this sequence with TG2b mRNA that binds more than a few bases in length. Thus, suppression of TG2b mRNA expression was not observed in these injection experiments.

On the other hands, the TG2-specific morpholino clearly inhibited TG2 mRNA expression. This sequence (5'-CATGAGCCGATGTCCAGAGAGCCAT-3') is complementary to exons 1 (35-57) of Danio rerio TG2b mRNA, which corresponds to the start site the cording region (35~, see below).

Danio rerio TG2b mRNA

1 CACCGCTGGA CCGCCGCAGC ATTACTTTAT AAACATGGCT CTGGACATCG GCTCATGGGA 60

|||||| |||||||||| |||||||

←TG2b MO binding region→

Based on your suggestion, we have added the following a description of the TG2b- specific morpholino to line 201 of the Results.

This TG2-specific morpholino has a complementary sequence to the start site of TG2b mRNA in the coding region of exon 1. Therefore, the entire expression of TG2b is downregulated (Supplement Figure 1).

-----------------------------------------------------------------------------------------

1-b) The authors at the very least should perform a PCR/gel based assay to demonstrate the incorrect splicing of the transcript after morpholino injection.

->We performed PCR using MO-treated samples and showed the results of gel electrophoresis in supplement figure 1. GAPDH was used as an internal control.

Compared to the Standard (Std) MO-treated sample, the expression of TG2b-MO-treated sample was clearly suppressed of TG2 mRNA.

Suppl. Figure 1

2) Retinoic acid signaling- while the authors show changes in expression of RA related genes after optic nerve injury, any role in regulating RGC survival or optic nerve regeneration in merely speculative. No manipulations of RA signaling were conducted, so it is difficult to state that these changes in expression contribute to downstream events such as the regulation of TG2 or HSF1. Thus the role of RA in this process is overstated throughout the manuscript.

->You are absolutely right. The relationship between retinoic acid-related gene expression and TG2 needs to be further confirmed. Therefore, we have revised the wording regarding RA signaling and TG expression in Discussion. (Please see Discussion)

3) Tg2 enzymatic assay (Figure 4)- No description of the assay is given. In the TG2 inhibitor panel of figure 4A, cells are not visible. This would be more convincing with a nuclear stain included, so we can see the cells in the inhibitor panel. Also, the picture is very zoomed in. It is difficult to tell what part of the eye is photographed. The RGC layer in adults is very thin (usually just a couple of cells layers thick) so this picture (control panel) is likely not of the ganglion cell layer. A picture with a sider field of view, with a counter nuclear stain should be added to the paper, along with a description of the enzymatic assay.

->A description of the measurement of in situ TG enzyme activity was added to Methods as “4.11 Measurement of in situ TG Enzyme Activity with flat mount retina”.

Retina was removed to make flat mount samples. FITC-labeled TG2-specific substrate were reacted with these unfixed flat-mounts retina using a 96 well microplate.

4) No negative control (no primary antibody control) was included for the immunohistochemistry. This is particularly important in the supplemental Figure, where the staining for all panels is weak. Some panels look a little darker than others, but it is difficult to interpret without the negative control.

-> We have added negative control (no primary antibody control) data to Supplement Figure 2.

5) The discussion is very brief, and just restates the results. Many papers from this field are not referenced, nor is there any description of how this work affects the field overall. Given the authors state that there is no increased HSF1 in the mouse retina after injury (supplemental Figure), but there is in the fish, differences with other animal models should be discussed.

 We have modified Discussion.

Kayo Sugitani, Ph.D.
--------------------------------------------------------
Department of Clinical Laboratory Science
Kanazawa University Graduate School of Medical Science
Kodatsuno 5-11-80, Kanazawa, 920-0942, Japan
Tel: +81-76-265-2599    Fax: +81-76-234-4369
--------------------------------------------------------

Reviewer 2 Report

Comments and Suggestions for Authors

In this paper, Kayo Sugitani et al. studied the functional roles and molecular mechanisms of Transglutaminase 2 (TG2) in the acute phase of fish optic nerve regeneration. Firstly, they found that the mRNA level of TG2 rapidly increased after optic nerve lesion (ONL). Then, they performed knockout experiments to demonstrate that HSF1 mRNA expression does not affect TG2 mRNA expression. Finally, they treated cells with a TG2-specific inhibitor and performed a ChIP assay to prove that TG2 directly regulates HSF1 by binding to its genomic region.

After reading through the manuscript, the following improvements are suggested:

  1. In Figure 2(d), the error bar is missing at 0.25 hrs.
  2. The authors did not show any phenotypes in the paper. Did you observe changes in the RGC survival rate after TG2 knockout or overexpression?
  3. The authors claimed that TG2 is regulated by the retinoic acid pathway. Please add the related citations in the manuscript.
  4. In Figure 4(a) lacks a control. No cells can be observed in the TG2 inhibitor panel.

Author Response

Dear Reviewer,

Thank you very much for your editorial work on our manuscript entitled “Transglutaminase 2 Regulates HSF1 Gene Expression in the Acute Phase of Fish Optic Nerve Regeneration” (Manuscript ID: ijms-3105568). 

We have revised it according to your comments and suggestions. The followings are the specific suggestions and our replies.

  1. In Figure 2(d), the error bar is missing at 0.25 hrs.

->We have corrected the error bar in Figure 2(d).

  1. The authors did not show any phenotypes in the paper. Did you observe changes in the RGC survival rate after TG2 knockout or overexpression?

->We performed an additional knockdown experiment by administering TG2-specific MO (Figure 7).

  1. The authors claimed that TG2 is regulated by the retinoic acid pathway. Please add the related citations in the manuscript.

-> We have added references related to the TG2 and retinoic acid signaling.

  1. In Figure 4(a) lacks a control. No cells can be observed in the TG2 inhibitor panel.

-> We have added a control (DMSO, 0 h) to Figure 4a.

Kayo Sugitani, Ph.D.
--------------------------------------------------------
Department of Clinical Laboratory Science
Kanazawa University Graduate School of Medical Science
Kodatsuno 5-11-80, Kanazawa, 920-0942, Japan
Tel: +81-76-265-2599    Fax: +81-76-234-4369
--------------------------------------------------------

Round 2

Reviewer 1 Report

Comments and Suggestions for Authors

The manuscript by Sugitani et al is much improved. The discussion and methods are more detailed. However, the use of a single morpholino to each gene to generate all data is problematic. Even with the additional details given, the use for a single morpholino to infer gene function is no longer the standard in this field.  I refer to the paper by Stanier et al., (PMID: 29049395) which highlights appropriate design and controls needed for the interpretation of morpholino based data. Given the ease that mutants can now be generated, F0 crispants or stable mutant lines should be generated to confirm the morpholino phenotype.   

Also, the description of the morpholino used indicates that it was designed to prevent tg2 expression. The additional details in this version of the manuscript shows that it binds in the 5’ UTR. Thus, as far as I can tell, the morpholino was designed to block translation of the message, not transcription of the gene. It is not expected that a translation blocking morpholino would affect mRNA levels of the gene unless there is a positive feedback loop, where by Tg2 protein positively regulates its own transcription. Then, blocking translation of the message could lead to a down-regulation of the mRNA levels as shown in the manuscript. Can the authors elaborate on the mechanism by which they believe this morpholino prevents expression of the gene?  

No information is given pertaining to the hsf1 morpholino.

Comments on the Quality of English Language

NA

Author Response

Dear Reviewer 1,

Thank you very much for your editorial work on our manuscript entitled “Transglutaminase 2 Regulates HSF1 Gene Expression in the Acute Phase of Fish Optic Nerve Regeneration” (Manuscript ID: ijms-3105568). 

We have revised it according to your comments and suggestions and thank you for your detailed peer review.

The followings are the specific suggestions and our replies.

1) The use of a single morpholino to each gene to generate all data is problematic. Even with the additional details given, the use for a single morpholino to infer gene function is no longer the standard in this field.  I refer to the paper by Stanier et al., (PMID: 29049395) which highlights appropriate design and controls needed for the interpretation of morpholino based data. Given the ease that mutants can now be generated, F0 crispants or stable mutant lines should be generated to confirm the morpholino phenotype.

-> We agree that an expression inhibition experiment using only one morpholino is not sufficient proof. This was mentioned as a point to be noted in the discussion. To compensate for this, we performed an experiment to suppress the expression of TG2 with a specific inhibitor,Z-DON. We need to explore this mechanism further using CRISPR-based gene editing, etc., as you have suggested. However, this was not possible in the limited time available for this 10-day project, we have modified Discussion. 

2) The description of the morpholino used indicates that it was designed to prevent tg2 expression. The additional details in this version of the manuscript shows that it binds in the 5’ UTR. Thus, as far as I can tell, the morpholino was designed to block translation of the message, not transcription of the gene. It is not expected that a translation blocking morpholino would affect mRNA levels of the gene unless there is a positive feedback loop, whereby Tg2 protein positively regulates its own transcription. Then, blocking translation of the message could lead to a down-regulation of the mRNA levels as shown in the manuscript. Can the authors elaborate on the mechanism by which they believe this morpholino prevents expression of the gene? 

-> The morpholino used in this experiment has the property of complementary binding to the last four bases of exon 1 and 19 bases of exon 2 (see below). We speculate that this MO may inhibit translation initiation in the cytoplasm and alter pre-mRNA splicing in the nucleus by targeting splice junctions or splice regulation sites. Indeed, in the present experimental system, TG2b mRNA expression was effectively suppressed.

transglutaminase 2b [Source:ZFIN;Acc:ZDB-GENE-030131-2576]

Exon/Intron

Length

Sequence  (Blue: Translated sequence)

1

222

GGTCGCTTTGGGAAAAATGTTTGCATAATTATTAAATGTCATATATAATAAAGAGTGATCTGCACGTATGTGATCGTCGGTCTGTCAGCTGTGAATGCGCATGTGGGCGTGTCGCTCTGCACAATAAAGACGCGCGTGCGTGTGAGAGCGCGCGCACACTGACTGTCACTGACTGAACGGAACTCACCGCTGGACCGCCGCAGCATTACTTTATAAACATGG

Intron 1-2

9,684

gtaagtgcgttttactctgcatcta..........aaattcttccgttgtgttgcttcag

2

180

CTCTGGACATCGGCTCATGGGATCTGGCGTGTAAATTTAATAACACGGACCACCACACGG AGCTGAACGGCACAGACCGGCTCATCGTGAGGAGAGGACAGGCCTTCACCATCAACCTGC AGCTCAACTCCGGCTCATACCAGCCCGGATACAGTCAGATCAACATCACTGCAGAGACTG

3) Minor editing of English language required

-> The English editing was done by MDPI's English editing service.

Kayo Sugitani, Ph.D.
--------------------------------------------------------
Department of Clinical Laboratory Science
Kanazawa University Graduate School of Medical Science
Kodatsuno 5-11-80, Kanazawa, 920-0942, Japan
Tel: +81-76-265-2599    Fax: +81-76-234-4369
-------------------------------------------------------

Round 3

Reviewer 1 Report

Comments and Suggestions for Authors

There response to my queries about the morpholino design are satisfactory.

The authors have responded to my concerns using a single morpholino, stating that the use of the TG2 inhibitor was used. While this is true, only the expression of HSF1 Was was shown using this inhibitor. Given a mutant analysis is not feasible in a short time frame, additional experiments are needed using the Z-DON inhibitor to replicate the morpholino phenotype (the apoptosis assay for example), as using a single morpholino to generate data in no longer the standard in the zebrafish field.